# Pexophagy: A Model for Selective Autophagy

**DOI:** 10.3390/ijms21020578

**Published:** 2020-01-16

**Authors:** Kyla Germain, Peter K. Kim

**Affiliations:** 1Cell Biology Program, Hospital for Sick Children, Toronto, ON M5G 0A4, Canada; kyla.germain@sickkids.ca; 2Department of Biochemistry, University of Toronto, Toronto, ON M5S 1A8, Canada

**Keywords:** selective autophagy, organelle quality control, peroxisomes, metabolism

## Abstract

The removal of damaged or superfluous organelles from the cytosol by selective autophagy is required to maintain organelle function, quality control and overall cellular homeostasis. Precisely how substrate selectivity is achieved, and how individual substrates are degraded during selective autophagy in response to both extracellular and intracellular cues is not well understood. The aim of this review is to highlight pexophagy, the autophagic degradation of peroxisomes, as a model for selective autophagy. Peroxisomes are dynamic organelles whose abundance is rapidly modulated in response to metabolic demands. Peroxisomes are routinely turned over by pexophagy for organelle quality control yet can also be degraded by pexophagy in response to external stimuli such as amino acid starvation or hypoxia. This review discusses the molecular machinery and regulatory mechanisms governing substrate selectivity during both quality-control pexophagy and pexophagy in response to external stimuli, in yeast and mammalian systems. We draw lessons from pexophagy to infer how the cell may coordinate the degradation of individual substrates by selective autophagy across different cellular cues.

## 1. Introduction to Peroxisomes

Peroxisomes are vital metabolic organelles found in nearly all eukaryotic cells. Peroxisomes were first described as “microbodies” in 1954 by the Swedish doctoral student Johannes Rodin, who observed spherical structures encompassing an electron-dense matrix in electron micrographs of mouse kidneys. In 1966, Christian de Duve and Pierre Baudhuin isolated microbodies from rat liver and using biochemical methods, found them to contain enzymes that both produce and decompose hydrogen peroxide, aptly renaming them peroxisomes [1].

Peroxisomes are small, single membrane-bound organelles ranging in size from 0.1 to 1 μm. Despite their high degree of variability amongst organisms and cell types, peroxisomes ubiquitously harbor a set of orthologous peroxin (PEX) proteins involved in peroxisome biogenesis and maintenance. The last half-century of research has identified peroxisomes as orchestrators of diverse metabolic functions including the alpha- and beta-oxidation of branched chain and very long chain fatty acids (VLCFA), respectively [2,3,4,5]. In addition to the synthesis of bile acids and ether lipids including plasmalogens, some evidence suggests that peroxisomes are the central site for the synthesis of the sterol precursor farnesyl pyrophosphate [6]. Through their metabolic activity peroxisomes produce a large quantity of reactive oxygen species (ROS) [7], yet concurrently harbor antioxidants to counteract ROS production such as the H_2_O_2_-decomposing enzyme catalase. Peroxisomes are promiscuous entities, which contact and cooperate with numerous organelles including the endoplasmic reticulum (ER) [8,9], mitochondria [10,11,12], lysosomes [13] and lipid droplets [14,15,16]. The significance of peroxisome biology is highlighted by their physiological roles in development, innate immunity and brain and liver function.

Recent reviews have been written about peroxisome biology and function across multiple organisms [17,18,19,20]. In response to mounting literature that emphasizes the importance of peroxisome homeostasis for cell health, the present review will discuss the maintenance of peroxisome homeostasis focusing primarily on peroxisome turnover by selective autophagy (pexophagy). We discuss the mechanisms of pexophagy in yeast and mammalian systems through the lens of selective autophagy to address pertinent questions in the fields of selective autophagy and organelle homeostasis.

## 2. Peroxisome Homeostasis

Owing to their necessity for cellular metabolism, peroxisome homeostasis is tightly regulated. Peroxisome homeostasis is achieved by balancing the rate of organelle formation and degradation to maintain the appropriate peroxisome abundance and quality. Peroxisome abundance is highly variable across tissue types, metabolic requirements and cellular stresses. At present, several metabolic signals are known to disrupt peroxisome homeostasis. These include changes in lipid and glucose levels, which induce an increase in peroxisome abundance through the activation of the peroxisome proliferator-activated receptor (PPAR) family of nuclear receptors [21]. Conversely, amino acid starvation or hypoxic conditions rapidly reduce peroxisome abundance by activating the selective degradation of peroxisomes through autophagy [5,6]. Decades of work exploring peroxisome homeostasis has culminated in a vast body of research detailing the mechanisms of peroxisome formation and degradation that allow peroxisome abundance to rapidly adapt to changes in environment.

Mammalian peroxisomes can be formed by two distinct but related pathways: de novo biogenesis from the ER, and growth and division of pre-existing organelles. The de novo biogenesis model suggests that immature peroxisomal vesicles emerge from a subdomain of the ER enriched in the peroxisomal membrane protein (PMP) PEX16 [22]. On the ER, PEX16 can recruit the PMP receptor PEX3 [23]. PEX3 together with the cytosolic PMP receptor PEX19, act to recruit PMPs containing membrane peroxisome targeting signals (mPTS) to the peroxisomal membrane [23,24]. ER-derived peroxisomal vesicles enriched in PMPs can later fuse together to become mature peroxisomes [25]. Recently it was shown that mitochondria-derived vesicles containing PEX3 can fuse with ER-derived vesicles [26], suggesting that peroxisomes can obtain membrane from diverse sources.

Peroxisomes can also form by growth and division, where pre-existing peroxisomes grow and elongate and eventually divide to form multiple peroxisomes. Mature peroxisomes elongate in a process dependent on PEX11 [27] and undergo asymmetric fission by the dynamin GTPase DRP1 and co-factors MFF and FIS1, forming two or more daughter organelles [24,25,26]. Proliferation of peroxisomes by growth and division is very rapid as exogenous expression of PEX11 in mammalian cells can double peroxisomes within the first 24 h [28]. Similarly, the activation of the PPAR family of nuclear receptors increases the expression of PEX11 family proteins that promote peroxisome elongation [29]. The precise mechanism by which these daughter peroxisomes obtain lipids is unknown, although evidence suggests that either vesicles or membrane contact sites with the ER can contribute to the increase in peroxisomal membrane bilayer [9,26,30].

Finally, peroxisomes must acquire matrix proteins to become functional compartments. As peroxisomes possess no genome, their nuclear encoded matrix proteins are imported post-translationally from the cytosol [31] (Figure 1). Matrix proteins are targeted to peroxisomes by virtue of two distinct peroxisome targeting signals (PTS): the C-terminal PTS1, and the less common N-terminal PTS2 [32,33]. PTS1 and PTS2-containing proteins are bound by the soluble receptors PEX5 [34] and PEX7 [35], respectively, that shuttle them to the peroxisome membrane through binding with membrane receptors PEX14 and PEX13 [36,37]. The interaction between PEX5 and PEX14 has been proposed to be sufficient to facilitate the deposition of PTS1-containing proteins into the peroxisome matrix [38]. Following dissociation from their cargo, soluble receptors are monoubiquitinated and recycled back to the cytosol by the AAA-type ATPase complex PEX1-PEX6-PEX26 [39,40,41,42]. Here, receptors are deubiquitinated to enable another round of import, or degraded by the ubiquitin proteasome system [43,44] (Figure 1).

Peroxisome quality control is itself regulated by a number of distinct mechanisms. Damaged or misfolded matrix proteins are degraded by the peroxisomal Lon protease 2 (LONP2), a homo-oligomeric ATP-dependent protease with chaperone-like activity [45,46]. LONP2 is necessary for peroxisome quality control as the loss of LONP2 results in degradation of the entire peroxisome [47]. Entire peroxisomes can be turned over either by autolysis or the lysosome via autophagy. 15-lipoxygenase-1 (ALOX15) mediates autolysis of peroxisome membranes that are damaged by peroxidation [48]. Moreover, peroxisomes are selectively degraded through autophagy in a process termed pexophagy. Studies using autophagy-incompetent mice revealed that during induction of peroxisome degradation, 20–30% of liver peroxisomes are degraded by LONP2 and ALOX15, and the remaining 70–80% are degraded via pexophagy [49]. Thus, pexophagy is the principal method of peroxisome degradation and will be the focus of this review.

## 3. Mechanisms of Macroautophagy

Autophagy (originating from the Greek phrase for self-eating) denotes a group of highly conserved catabolic pathways that transport intracellular components to the vacuole or lysosome for their degradation [50,51]. Basal autophagy degrades cytosolic debris including long-lived proteins or superfluous organelles, contrary to the ubiquitin proteasome system (UPS), which digests predominantly short-lived proteins [50,51,52,53]. Furthermore, autophagy can be induced by environmental stimuli such as ROS, hypoxia or starvation, to generate metabolites that can be used for de novo biomolecule synthesis or as substrates for energy production [50,54,55]. Autophagy plays essential roles in numerous biological processes such as programmed cell death, inflammation and innate immunity, and its disruption contributes to cancer, obesity and neurodegenerative disease [56].

Three forms of autophagy have been described: chaperone mediated autophagy (CMA), microautophagy and macroautophagy. In CMA, proteins containing a KFERQ motif are shuttled to the lysosome by HSC70 and translocated across the membrane by the lysosome-associated membrane protein 2A (LAMP2A) for degradation [57]. In microautophagy, the vacuole membrane directly invaginates to engulf adjacent cytosolic components. Microautophagy has been well described in yeast [58], but remains scantly explored in mammals [59]. Macroautophagy involves the sequestration of cytosolic components inside of a double-membraned vesicle called the autophagosome that itself fuses with the vacuole/lysosome [50]. Macroautophagy, hereafter referred to as autophagy, is the most prominent form of autophagy occurring in mammalian cells [60].

Much of the present day understanding of autophagy is owed to studies carried out in yeast. To date, 36 autophagy-related genes (*ATG*s) have been identified, and constitute the core machinery that executes autophagy [51] (Table 1). The process of autophagy can be divided into five steps: initiation, nucleation, elongation, sequestration and degradation (Figure 2A). Autophagy initiation refers to the formation of the initial pre-autophagosomal structure (PAS) in yeast, or isolation membrane in mammals. Autophagy initiation relies on the activity of the Atg1p kinase in yeast, or ULK isoforms 1 and 2 (ULK1/2) in mammals, which assemble in a complex to initiate autophagosome biogenesis [61,62,63,64,65,66,67,68]. (Note herein that the yeast protein will be named followed by ‘/’ the mammalian protein homologue, and ULK1 will represent ULK1/2 isoforms.) The source of the mammalian autophagosomal membrane is debated, with evidence to suggest that the ER, mitochondria, plasma membrane and endosomal membrane may contribute [69,70,71,72,73]. Nucleation describes the recruitment of specific molecules to the PAS/isolation membrane that facilitate elongation to produce an autophagosome [60]. Nucleation is driven by the class III phosphatidylinositol (PtdIns) 3-kinase (PI3K) Vps34p/VPS34 complex [74,75], which produces autophagosome-specific PtdIns 3-phosphates and recruits necessary effector proteins [76]. Autophagosome elongation is driven by two ubiquitin-like conjugation systems. The first results in formation of the Atg12p-Atg5p-Atg16p/ATG12-ATG5-ATG16 complex [77,78,79], and the second results in the lipidation of Atg8p/LC3-I to produce Atg8p/LC3-II-phosphatidylethanolamine (PE) [80,81,82,83,84,85]. This lipidation event enables Atg8p/LC3-II to interact with the autophagosomal membrane and facilitates cargo capture inside of the maturing autophagosome [86]. Elongation is furthered by Atg9p/ATG9-containing vesicles that deliver additional membrane to growing autophagosomes [87]. As the autophagosome elongates, it sequesters cytosolic material to be degraded. Autophagosome closure and fusion with the lytic compartment occurs in a process dependent on RAB GTPases and SNARE proteins [88,89,90,91,92]. In yeasts autophagosomes form adjacent to the vacuole and the ER, thus vacuolar fusion and degradation occur immediately following autophagosome closure [93,94]. In mammals autophagosomes are transported on microtubules towards the microtubule organizing centre where lysosomes are concentrated [95,96,97,98]. Here, autophagosomes fuse with lysosomes whose acidic hydrolases degrade their contents and recycle the core components back to the cell [97].

## 4. Selective Autophagy Overview

Autophagy was long perceived as a non-specific process, wherein random portions of cytoplasm containing various cellular components are sequestered inside of autophagosomes and degraded following autophagy induction. However, a study investigating global protein dynamics during amino acid starvation found that there appears to be ordered degradation of cellular components during 36-h amino acid starvation-induced autophagy [119]. Using an unbiased quantitative mass spectrometry proteomics approach, the authors observed that cytosolic proteins and proteasomes are degraded first, followed by ribosomes and finally organelles like the ER. Subsequent studies have demonstrated that mitochondria are protected from autophagic degradation during amino acid starvation by forming hyper-fused networks that are unaccommodating to sequestration within autophagosomes [120,121]. These findings suggest that selectivity can be achieved during starvation-induced autophagy to allow the ordered degradation of distinct cellular components.

Selective autophagy is the degradation of specific cellular components, or “cargoes,” through the autophagy pathway [122]. The process of selective autophagy can be separated into three steps: designation, targeting and sequestration and degradation (Figure 2B). Designation involves the addition or recruitment of a molecular tag to distinct cargoes for their recognition by the autophagy machinery. Targeting and sequestration occur when designated cargoes are targeted to nascent autophagosomes by specific receptors, allowing the autophagosome to elongate around and sequester the cargo. Degradation of cargoes ensues following the fusion of the autophagosome with a vacuole/lysosome.

Cargoes are designated for autophagy by various mechanisms. In mammalian cells ubiquitination commonly serves as a molecular “eat me” signal for the core autophagic machinery [111,123]. Notably, designation signals including ubiquitination can be dissipated from cargoes by regulatory components to avert autophagy. Selectivity is achieved by high affinity interactions between autophagy receptors and designated cargoes [123,124]. Autophagy receptors recognize and bind designated cargoes through ubiquitin-binding domains or alternate motifs. Concomitantly, autophagy receptors interact with Atg8p/LC3 on the autophagosome via evolutionarily conserved motifs—Atg8p-interacting motif (AIM) in yeast, and LC3-interacting region (LIR) or ubiquitin-interacting-motif (UIM) in mammals [125]. Autophagy receptors hereby target designated cargoes to the core autophagy machinery and facilitate their sequestration inside of autophagosomes. Numerous autophagy receptors have been identified in mammals including: SQSTM1 or p62, NBR1, OPTN, NDP52, TAX1BP and NIX or BNIP3L, while three have been identified in yeast: Atg30p, Atg33p and Atg36p [123]. Autophagy receptors are degraded alongside their bound cargo upon fusion with the lytic compartment.

More recently, it has been shown that some autophagy receptors can act upstream of cargo targeting and sequestration by directly interacting with autophagy initiation machinery. Both NDP52 and OPTN were shown to interact with and recruit the ULK1-initiation complex to damaged mitochondria preceding their designation [126,127]. It was further shown that NDP52 and p62 interact with and recruit the ULK1 complex to cytosol-invading bacteria and insoluble protein aggregates, respectively, to promote their degradation [128,129]. These findings, reviewed in [130], suggest that autophagy receptors can recruit the autophagy initiation machinery directly to cargoes to promote autophagosome biogenesis around the cargo, rather than targeting designated cargoes to growing autophagosomes.

Various cellular components have been identified as selective autophagy cargoes, including protein aggregates (aggrephagy), organelles such as peroxisomes (pexophagy), mitochondria (mitophagy), ER (reticulophagy), lipid droplets (lipophagy), as well as bacteria (xenophagy) and viruses (virophagy). The designation, targeting and sequestration of distinct selective autophagy cargoes are mediated by unique receptor and regulator proteins. In this way, selective autophagy can act as a highly specific quality control system to monitor cellular components and designate superfluous or damaged components for autophagy.

Perhaps the most well studied example of selective autophagy is the PTEN-induced kinase 1 (PINK1)-Parkin mitophagy pathway [131]. Functional mitochondria import PINK1 into the inner mitochondrial membrane where it is quickly proteolytically degraded [132]. Dysfunctional, import-incompetent mitochondria undergo a resultant buildup of PINK1 on their outer membrane, where it exerts its kinase activity and phosphorylates nearby proteins, including ubiquitin and Parkin [133]. Phospho-ubiquitin acts as a signal to recruit mitophagy factors including the autophagy receptors, OPTN and NDP52 [127]. Meanwhile, Parkin is itself an E3 ubiquitin ligase and following phospho-activation, acts to amplify the mitophagy signal by ubiquitinating mitochondrial outer membrane proteins, effectively designating mitochondria for mitophagy [134,135]. Designated mitochondria are targeted and sequestered within autophagosomes by OPTN and NDP52 and undergo subsequent lysosomal degradation. Thus, the import-competency of mitochondria acts as a meter for its quality control allowing the selective targeting of dysfunctional mitochondria for mitophagy.

An outstanding conundrum is how environment-induced autophagy and quality-control autophagy cooperate. Here, we defined environment-induced autophagy as autophagy resulting from a change in environment or extracellular stimuli such as starvation or hypoxia. We defined quality-control autophagy as the targeting and sequestration of specific components for autophagy based on functionality, such as PINK1-Parkin mitophagy. Whether environment-induced autophagy is sometimes non-selective or exclusively selective remains a topic of discussion. Furthermore, how the cell regulates selectivity during environment-induced autophagy is unclear. Here, we outlined why pexophagy is an excellent model for both environment-induced and quality-control autophagy and discuss what we could infer about selective autophagy from pexophagy.

## 5. Pexophagy: The Selective Autophagic Degradation of Peroxisomes

Environment-induced pexophagy was first examined following a shift in nutrient availability. Yeasts can utilize different carbon sources for energy, so when grown in media reliant on peroxisome metabolism they will rapidly increase peroxisome abundance. For example, treating *S. cerevisiae* with oleic acid [136], or *P. pastoris* and *H. polymorpha* with methanol, oleate or amines, induces peroxisome biogenesis and the formation of giant peroxisome clusters [137]. Contrarily, shifting yeasts from peroxisome-dependent carbon sources to peroxisome-independent carbon sources such as glucose triggers their immediate degradation via pexophagy. Similarly, treating rodents with peroxisome proliferating stimuli (phthalate esters and hypolipidemic drugs) rapidly increases peroxisome abundance, whereby upon removal of the stimulus large-scale pexophagy is triggered [138,139,140]. Peroxisome degradation is inhibited with treatment of substrates of peroxisomal fatty acid oxidation, long and very-long-chain fatty acids (>C16:0), but not with substrates of mitochondria fatty acid oxidation [21], providing compelling evidence for the selectivity of peroxisomal sequestration within autophagosomes during environment-induced autophagy.

Mammalian pexophagy plays a critical role in peroxisome quality control. Mammalian peroxisomes have half-lives of 1.5–2 days, highlighting the dynamicity of their basal turnover [141,142,143]. One common theme that has emerged in yeasts and mammals is the necessity of the matrix import machinery to maintain peroxisome quality, wherein defects in matrix import machinery act as a signal to designate peroxisomes for pexophagy. Below, we discuss our knowledge of the mechanisms and regulation of peroxisome designation, targeting and sequestration, and degradation by autophagy in yeast and mammalian systems, highlighting both environment-induced pexophagy and quality-control pexophagy.

### 5.1. Peroxisome Designation for Degradation in Yeast

The specific mechanisms of designating peroxisomes for pexophagy differ between yeast and mammalian cells. However, for both systems, the inability to import matrix proteins can designate peroxisomes for quality-control pexophagy. One example is in *S. cerevisiae* where the loss of peroxisomal AAA-type ATPase (Pex1p-Pex6p-Pex15p) renders peroxisomes import-incompetent, and promotes enhanced pexophagy [99]. In cells lacking components of the AAA-type ATPase, peroxisomes accumulate ubiquitinated Pex5p on their outer membrane and are degraded in a process dependent on the *S. cerevisiae* autophagy receptor for pexophagy (pexophagy receptor). Intriguingly, their degradation was shown to be ubiquitin-independent, suggesting that ubiquitination is not a signal for pexophagy in yeast, and an alternate signal exists to designate peroxisomes for autophagy.

The pexophagy receptors identified in yeasts are Atg30p in *P. pastoris* [108] and Atg36p in *S. cerevisiae* [109,110], which share functional similarities but are not structural homologs (Figure 3A). Atg30p and Atg36p are classified as *bona fide* pexophagy receptors because in addition to peroxisome binding sites, they contain an AIM to interact with Atg8p and when over-expressed they promote pexophagy [114]. Both pexophagy receptors are constitutively localized to peroxisomes through interactions with peroxisome membrane proteins, yet only become activate pexophagy receptors following phosphorylation. Phosphorylation of peroxisome bound-Atg30p and -Atg36p at specific residues allows them to interact with core autophagy machinery and facilitates peroxisome targeting and sequestration within autophagosomes [108,110,113]. In this way, the phosphorylation of yeast pexophagy receptors acts as the molecular tag that designates peroxisomes for degradation.

In *P. pastoris* the pexophagy receptor Atg30p interacts with Pex3p [144] and the acyl-CoA binding protein, Atg37p, on the peroxisome [145]. Atg30p contains one Atg37p-binding site in its middle domain and two Pex3p-binding sites, one of which is mutually exclusive with Atg37p-binding. Atg30p is recruited to peroxisomes through interactions with the C-terminal cytosolic domain of Pex3p [144]. Atg30p in turn recruits Atg37p to the pexophagic receptor protein complex (RPC), where Atg37p promotes Atg30p-binding to autophagy machinery. Interestingly, Pex3p and Atg37p depend on each other for their correct localization, as Atg37p is recruited to peroxisomes in a Pex3-dependent manner, and conversely the lack of Atg37p mislocalizes Pex3p [113,146].

Atg30p undergoes two phosphorylation events that designate peroxisomes for pexophagy and recruit autophagy machinery [113,114]. Atg30p is phosphorylated at S71 by an unknown kinase, and at S112 by Hrr25p (Figure 3A) [113,114]. Hrr25p-Atg30p association is inhibited when Pex3p is bound to Atg30p’s middle domain, however Atg37p recruitment to the RPC displaces Pex3p and allows Hrr25p recruitment and Atg30p phosphorylation [113,146]. Pex3p re-binding to Atg30p’s middle domain is thought to promote Hrr25p dissociation, which could promote pexophagy termination. Thus, in *P. pastoris* peroxisomes are designated for pexophagy through the actions of Pex3p and Atg37p on the peroxisome, which regulate the phospho-activation of the pexophagy receptor Atg30p. Phospho-activated Atg30p is able to bind core autophagic machinery to promote targeting and sequestration of peroxisomes within autophagosomes.

Similarly, the *S. cerevisiae* pexophagy receptor Atg36p is tethered to peroxisomes by Pex3p-binding and is phosphorylated by Hrr25p at S97 (corresponding to Atg30p S112) to promote pexophagy (Figure 3A) [109,110]. Interestingly, Atg36p is phosphorylated under both peroxisome proliferating and pexophagy-inducing conditions, however the degree of phosphorylation drastically increases during pexophagy induction resulting in increased interactions with autophagy machinery. Currently, the regulatory mechanisms governing Atg30p and Atg36p phosphorylation at the peroxisome are not known. Whether Atg30p and Atg36p are differentially phosphorylated to designate specific, import-incompetent peroxisomes for quality-control pexophagy requires further investigation.

The mechanism of peroxisome designation for degradation may differ in *H. polymorpha.* In *H. polymorpha*, pexophagy is abolished following short N-terminal deletions of Pex14p that inhibit its ability to bind matrix import machinery, implying a role for the matrix import cycle in regulating pexophagy [99]. Additional studies in *H. polymorpha* report that peroxisomes in which Pex3p is abruptly removed are rapidly degraded even in peroxisome-proliferating conditions by the UPS and degraded prior to pexophagy [105]. This is contrary to the role for Pex3p in *P. pastoris* and *S. cerevisiae* pexophagy where Pex3p is required for the recruitment of pexophagy receptors to peroxisomes. This suggests that in *H. polymorpha*, Pex3p either directly prevents peroxisome designation for pexophagy, or that its loss results in peroxisome dysfunction that promotes pexophagy [147].

### 5.2. Peroxisome Designation for Degradation in Mammals

In mammalian cells ubiquitin acts as a designation signal for selective autophagy [148]. It was first shown that mono-ubiquitination of a long lived cytosolic protein or the cytosolic face of an organelle can induce their autophagic degradation [111]. For example, expressing PEX3 or PMP34 fused to a ubiquitin motif on the cytosolic face of the peroxisomal membrane was sufficient for peroxisomes to be recognized by the autophagy receptor p62 and targeted to autophagosomes for degradation [111]. Further work by Deosaran, E. et al. supported the role for ubiquitin as a designation signal for pexophagy by demonstrating that autophagy receptors NBR1 and p62 bind peroxisomes through their ubiquitin associating (UBA) domains, and postulated PEX5 as a potential ubiquitinated signal for pexophagy [112].

Direct evidence connecting ubiquitin designation of peroxisomes to quality-control pexophagy did not arise until 2015, when Nordgren, M. et al. furthered the link to the matrix protein import cycle [102]. Noting that PEX5 receptor recycling is a meter of peroxisome import-competency, they hypothesized that disrupting receptor recycling would induce pexophagy. Indeed, they found that over-expressing PEX5 with a C-terminal EGFP tag in SV40 large T antigen-transformed mouse embryonic fibroblasts (MEFs) results in the retention and mono-ubiquitination of PEX5-EGFP at the peroxisome followed by pexophagy. Ubiquitinated PEX5 was also shown to mediate pexophagy in cells exposed to exogenous oxidative stress. During oxidative stress PEX5 has been shown to bind and target ataxia-telangiectasia mutated (ATM) to peroxisomes, where ATM phosphorylates PEX5, which promotes its ubiquitination and subsequent targeting to autophagosomes [149].

Ubiquitinated PEX5 was also shown to accumulate on import-incompetent peroxisomes due to mutations in any of the three genes that form the peroxisomal AAA-type ATPase: *PEX*1, *PEX*6 and *PEX2*6. Loss of the AAA-type ATPase function in human fibroblast and HeLa cells results in defective receptor recycling and the accumulation of ubiquitinated PEX5 on peroxisomal membranes, concomitant with peroxisome loss [100]. This peroxisome loss was rescued by pharmacologically or genetically inhibiting pexophagy. These findings imply a peroxisomal quality control mechanism wherein peroxisomes defective in receptor recycling undergo a build-up of ubiquitinated PEX5, which specifically designates import-incompetent peroxisomes for pexophagy (Figure 3B).

Evidence for the ubiquitination of peroxisomal proteins as a signal for autophagic degradation was further supported by the identification of an E3 ubiquitin ligase that induces pexophagy [101]. PEX2 is a component of the peroxisomal E3 ubiquitin ligase complex composed of three RING peroxins, PEX2-PEX10-PEX12, that is required for PEX5 receptor recycling [150]. RNA interference depletion studies combined with overexpression studies found that PEX2 was necessary and sufficient to ubiquitinate peroxisomes for their recognition by pexophagy receptors in basal conditions (Figure 3B). Furthermore, PEX2 depletion abrogated amino-acid starvation-induced pexophagy in cultured cells. Importantly, PEX2 was found to ubiquitinate both PEX5 and PMP70 [101]. Whether PEX5 and PMP70 are the only proteins selectively ubiquitinated is not known, however, given the non-selectivity of other autophagy E3 ubiquitin ligases, PEX2 likely ubiquitinates additional peroxisomal membrane proteins during amino acid starvation. It was further shown that PEX2 levels were upregulated during amino-acid starvation via mTORC1 inhibition in both cultured cells and in the livers of protein-starved mice [101,151]. These findings suggest a putative mechanism of selective peroxisome degradation during amino acid starvation, where PEX2 expression is upregulated resulting in mass peroxisome ubiquitin-designation and degradation. This study also suggests a role for PEX2 in quality control-pexophagy; PEX2 ubiquitinates PEX5 and upon defective receptor recycling ubiquitinated PEX5 accumulates at the peroxisome membrane and signals for pexophagy. Thus, the same machinery may be required for peroxisome designation during both environment-induced and quality-control pexophagy, while the events leading up to peroxisome designation remain distinct.

Opposing the action of PEX2 on peroxisomes is the deubiquitinating enzyme USP30 (Figure 3B). First described as a mitophagy regulator [152], USP30 has been shown independently by two groups to regulate pexophagy by targeting to peroxisomes [103,104]. Marcassa, E., et al. found that an endogenous portion of USP30 localizes to peroxisomes via an N-terminal transmembrane domain where it acts to reduce basal pexophagy [103]. Riccio, V., et al. subsequently reported that over-expression of USP30 prevents amino-acid starvation-induced pexophagy, by counteracting PEX2-mediated ubiquitination of PEX5 and PMP70 [104]. These studies highlight the importance of ubiquitin designation for pexophagy and demonstrate that dissipation of ubiquitin from peroxisomes serves to counteract pexophagy basally and during amino acid starvation.

There is some evidence for a ubiquitin-independent pathway that designates peroxisomes for pexophagy. One of the first studies examining the designation of mammalian peroxisomes for degradation was performed in Chinese hamster ovary (CHO) cells where peroxisomes were shown to be preferentially degraded over cytosolic proteins after amino-acid starvation and recultivation in nutrient-rich media [107]. During starvation, the authors demonstrate that LC3-II out competes PEX5 for binding to the N-terminal region of PEX14, ultimately targeting peroxisomes to autophagosomes (Figure 3B). They postulate that import-competent peroxisomes deter pexophagy through PEX14-PEX5 binding, whereas import-incompetency frees PEX14 allowing it to bind LC3-II and facilitate pexophagy. In this model, peroxisomes are not designated for pexophagy with ubiquitin, and instead the availability of PEX14 to bind autophagic machinery determines whether pexophagy occurs.

The same group also demonstrated that PEX3 might also act to designate peroxisomes for pexophagy. They showed that over-expression of PEX3 results in ubiquitinated, clustered peroxisomes that are sequestered inside of autophagosomes (Figure 3B) [106]. Furthermore, PEX3-induced pexophagy was inhibited by depleting pexophagy receptors or pharmacologically inhibiting pexophagy. Quizzically, PEX3 ubiquitination was not required for pexophagy indicating that PEX3 induces the ubiquitin designation of peroxisomes through a yet unidentified mechanism but is not itself a target of ubiquitination. However, it should be considered that over-expression of PEX3 leads to its mitochondrial localization, which could impact pexophagy through mitochondria-peroxisome interactions.

The existence of complex ubiquitin-dependent and -independent designation mechanisms in mammals highlights the importance of tagging distinct peroxisomes for degradation. Yeast pexophagy receptors directly bind to peroxisomal proteins and are phospho-activated on peroxisomes to designate them for pexophagy. It is unclear whether yeast pexophagy receptors can distinguish between peroxisomes, or if they solely respond to cellular cues that promote their phosphorylation and mass peroxisome degradation. It remains possible that there are undiscovered regulatory mechanisms governing Pex3p and Atg37p activity at the peroxisome that regulate the phospho-activation of Atg30p and Atg36p, which could confer the ability to designate specific peroxisomes. It is equally possible that yeasts are unable to selectively target individual peroxisomes within a population for pexophagy. One explanation for this discrepancy between yeast and mammalian pexophagy could be the difference in peroxisome abundance between the two systems. While yeast cells have 1–20 peroxisomes dependent on the species, mammalian cells can harbor anywhere from several hundred to a thousand peroxisomes [153]. As such, the mammalian cell requires an energy-efficient mechanism to monitor peroxisomes for quality control and specifically degrade dysfunctional peroxisomes while protecting the integrity of the larger peroxisomal network. Whereas, the lower number of peroxisomes in yeast cells could make it unnecessary to monitor individual peroxisomes for quality control.

### 5.3. Peroxisome Targeting and Sequestration

Peroxisomes designated for degradation are targeted to the phagophore assembly site (PAS) in yeast, or phagophore in mammals, to be sequestered within autophagosomes. PAS formation in yeasts is mediated by the interactions between autophagy receptors and the core autophagy machinery. The pexophagic PAS is organized by pexophagy receptors as well as Atg11p and Atg17p [154] (Figure 3A). Atg11p is a common ‘selectivity factor’ shared by various selective autophagy pathways that interacts with autophagy receptors and connects them to the core autophagic machinery including Atg1p and Atg17p, a scaffolding protein [154].

Designated peroxisomes are targeted to PAS in yeast and sequestered within autophagosomes through interactions with phosphorylated pexophagy receptors. In *P. pastoris*, Atg30p interacts with Atg11p via an Atg11p-binding region (A11-BR) and Atg8p via an AIM [114]. Phosphorylation of Atg30p’s A11-BR (S112) promotes Atg11p recruitment and binding, whereas phosphorylation of Atg30p’s AIM (S71) promotes Atg8p binding. Likewise, phosphorylated Atg36p in *S. cerevisiae* recruits and binds Atg11p and Atg8p [102,149]. These high-affinity interactions of Atg30p and Atg36p with Atg11p and Atg8p target and tether designated peroxisomes to the PAS, facilitating sequestration of peroxisomes within the growing autophagosome (Figure 3A).

In yeasts the peroxisomal fission machinery is required to aid peroxisome sequestration within autophagosomes (Figure 3A). Dnm1p and vacuolar protein sorting-associated protein 1 (Vps1p) are recruited to peroxisomes via interactions with Atg11p and Atg36p to constrict peroxisomes for pexophagy [118]. In *H. polymorpha*, the segregation of a mutant form of aggregated catalase was observed by asymmetric peroxisome fission directly coupled to pexophagy in a process dependent on Dnm1p, Pex11p, Atg1p and Atg11p [115,116]. This suggests a mechanism of organelle quality control whereby peroxisomes may concentrate damaged lumenal proteins within a portion of peroxisome that is pinched off by asymmetric fission, followed by immediate degradation through coupling to the pexophagic machinery. It is not known whether fission is similarly required for mammalian pexophagy. However, as peroxisomes in mammalian cells are significantly smaller than in methylotrophic yeasts [155] it is possible that fission is not a requisite for peroxisome sequestration within autophagosomes.

Atg30p and Atg36p have no mammalian homologs, perhaps owing to the requirement for ubiquitin in designating peroxisomes for mammalian pexophagy. Mammalian peroxisomes are instead targeted to phagophores to be sequestrated by two ubiquitin binding proteins, p62 and NBR1 [106,112]. Over-expression of either p62 or NBR1 promotes pexophagy, whilst their depletion reduces basal pexophagy [111]. p62 and NBR1 possess similar domain architecture, each comprising a coiled-coil domain that promotes their homo- and hetero-oligomerization, a LIR that facilitates interaction with LC3-II and a UBA that mediates binding of ubiquitin-designated peroxisomes. NBR1 uniquely contains an amphipathic α-helical J domain that recognizes phosphatidylinositol-phosphates (PIPs) and phosphatidic acid lipids, conferring selectivity of NBR1 to both late endosomes and peroxisomes [112]. The coincident binding of NBR1’s J-domain and the UBA achieves its specific localization to designated peroxisomes. Both autophagy receptors also contain a PB1 domain that allows them to bind to each other. The interaction of p62 with NBR1 has been proposed to cluster peroxisomes and cooperate with NBR1 to enhance pexophagy efficiency, but is not required when NBR1 is in excess [112]. In this way, NBR1 is the primary pexophagy receptor while p62 acts to enhance NBR1-mediated pexophagy. It is also possible that rather than targeting ubiquitinated peroxisomes to autophagosomes, p62 and NBR1 may instead recruit the autophagy initiation machinery to peroxisomes in a similar mechanism as shown for mitochondria and bacteria. In this model, p62 and NBR1 would interact with and recruit the ULK1 complex to initiate autophagosome biogenesis around peroxisomes to encapsulate them. As overexpression of NBR1 has been shown to cluster peroxisomes [112], targeting of the autophagy initiation machinery to peroxisome clusters would allow for efficient degradation of peroxisomes.

Unlike Atg30p and Atg36p, it is not known whether NBR1 and p62 require phospho-activation to enact pexophagy. Interestingly, both NBR1 and p62 are also receptors for aggrephagy, the autophagic degradation of ubiquitinated protein aggregates [156,157,158]. NBR1 and p62 cooperate to cluster ubiquitinated proteins into larger aggregates and aid their targeting and sequestration within autophagosomes through interactions with LC3-II. Post translational modifications including phosphorylation of NBR1 and p62 have been studied in the context of aggrephagy. In response to proteotoxic stress, p62 is phosphorylated in its UBA at S409 by ULK1 [159] or at S403 by casein kinase 2 (CK2) [160], which increases its affinity for ubiquitin and promotes aggrephagy activity. Furthermore, phosphorylation of p62 at S351 by an unknown kinase activates a positive-feedback loop, which drives p62 expression via the Keap1-Nrf2 cytoprotective pathway [161]. Notably, phosphorylation of NBR1 at Thr586 by GSK3 [162] has been shown to negatively regulate aggrephagy activity. As phosphorylation of NBR1 and p62 has been shown to regulate aggrephagy, it is possible that their phosphorylation also plays a role in regulating pexophagy and warrants further investigation.

Pexophagy receptors are essential to target peroxisomes to the forming autophagosome and achieve sequestration in both yeast and mammalian pexophagy. While yeast pexophagy receptors bind to all peroxisomes, mammalian pexophagy receptors exclusively bind to ubiquitinated peroxisomes to initiate pexophagy. In this way, they can constantly scan the cell’s peroxisome population for ubiquitinated peroxisomes to be degraded, highlighting the capacity of mammalian cells to enact both environment-induced and quality-control pexophagy. Finally, the necessity of fission machinery during mammalian pexophagy has not been shown, perhaps owing to the smaller size of mammalian peroxisomes.

### 5.4. Peroxisome Degradation

Peroxisomes sequestered within autophagosomes are delivered to the lytic compartment and degraded in the same manner as other autophagy cargoes. In yeasts, a PtdIns3P-binding protein, Atg24p, is required for vacuolar fusion during pexophagy, but not for general autophagy in *P. pastoris* and *S. cerevisiae* [117]. In mammals, no pexophagy-specific fusion or degradation factors have been described. Furthermore, there are no mechanisms distinguishing environment-induced pexophagy from quality-control pexophagy at this late stage in autophagic degradation.

One persisting question in the field of pexophagy regarded the fate of cytosolic peroxisomal proteins during environment-induced pexophagy, such as soluble receptors PEX5 and PEX7, which are 70–90% cytosolic [118]. To address this, a study in *P. pastoris* reported that during pexophagy induction by nutrient shift, cargo-bound Pex5p and Pex7p are degraded by an Atg30p-independent, yet Atg11p-dependent selective autophagy pathway [118]. The authors proposed that this was a protective mechanism elicited by the yeast cells to prevent toxicity from cytosolic peroxisomal enzymes during pexophagy induction. How this degradative pathway is regulated and whether a similar mechanism exists in mammals remains to be elucidated.

## 6. Regulation of Pexophagy

Yeast pexophagy is regulated through glucose-sensing and mitogen-activated protein kinase (MAPK) cascades, although the precise mechanisms of how they affect pexophagy and whether the two signaling pathways are related is not fully understood. Given that yeast pexophagy can be induced by a shift from growth in a non-fermentable carbon source to glucose, the existence of glucose-sensing pathways regulating yeast pexophagy is unsurprising. In *S. cerevisiae*, the cell-surface glucose sensor composed of the G-protein-coupled receptor Gpr1p and G-protein Gpa2p, which are involved in cAMP-signaling, were shown to regulate glucose-induced pexophagy [163]. Additionally, the high and low affinity glucose sensors Snf3p and Rgt2p, respectively, are reported to play a role in glucose-sensing to initiate pexophagy [164]. Homologs of Rgt2p and Snf3p function similarly in *P. pastoris* (Gss1p) [163] and *H. polymorpha* (Gcr1p) [165]. These proteins and their relevant signaling cascades provide a connection from nutrient-sensing to pexophagy induction.

Additional studies have reported a role for the MAPK, Slt2p, and several upstream components of this signaling pathway in regulating *S. cerevisiae* pexophagy [166]. Slt2p is necessary for pexophagy, but not the sequestration of peroxisomes within autophagosomes, suggesting a role in late stage pexophagy such as vacuolar fusion and degradation [166]. More recently, Slt2p was found to be necessary for mitophagy, but no other forms of autophagy [167], raising the possibility that the regulation of yeast pexophagy and mitophagy may converge at the vacuole.

Finally, a role for palmitoyl-CoA in yeast pexophagy regulation has been suggested. In vitro work suggests that palmitoyl-CoA and Atg30p compete for the same binding site on the N-terminal cytosolic domain of Atg37p, although it does not affect Atg37p-Pex3p binding [145]. These findings prompt speculation as to whether increasing concentrations of palmitoyl-CoA may inhibit pexophagy in certain conditions. Importantly, Atg37p-palmitoyl-CoA binding is required to first localize Atg37p to peroxisomes before it can act to mediate pexophagy. The exact role of palmitoyl-CoA in physiological pexophagy and whether it also plays a role in mammalian systems remain unclear. However, given that palmitoyl-CoA is unlikely metabolized in mammalian peroxisomes, it may be a specific regulatory pathway in yeast.

In mammalian cells, several extracellular and intracellular stimuli can activate pexophagy including amino acid starvation, hypoxia and ROS (Figure 4). Amino acid starvation-induced pexophagy was shown to be mediated by the E3 ubiquitin ligase PEX2 though the inhibition of mTORC1 [101]. In basal conditions, PEX2 is rapidly turned over resulting in overall low expression of PEX2. However, PEX2 protein levels were stabilized in vitro and in vivo during both amino acid starvation and Rapamycin treatment, suggesting that the mTORC1 pathway may regulate PEX2 stability. Importantly, PEX2 expression was biphasic during amino acid starvation and Rapamycin treatment, increasing during the first 2 h before returning to basal expression levels [101]. As such, it is tempting to speculate that during basal conditions mTORC1 promotes the proteasomal degradation of PEX2. Whereas, upon starvation or other conditions that inhibit mTORC1, PEX2 is stabilized and can promote the ubiquitination of peroxisomal proteins to designate peroxisomes for pexophagy.

Peroxisomes are dependent on molecular oxygen for their functionality in oxidative metabolism. In 2014 Walter, M. et al. [168] elucidated a regulatory pathway controlling peroxisome abundance in response to oxygen sensing [169]. The transcription factors hypoxia inducible factor alpha 1 and 2 (HIF-1a and HIF-2a), are master regulators of the cellular adaptive response to hypoxia. In oxygen-rich conditions, HIF-1a and HIF-2a are targeted for proteasomal degradation by the Von Hippel-Lindau (VHL) protein. Walter, M. et al. examined the role of HIF-2a signaling in pexophagy, and found that livers of *VHL*^−/−^/*HIF*-*1a*^−/−^ mice with active HIF-2a signaling had fewer peroxisomes due to NBR1- and p62-mediated pexophagy, compared to livers of *VHL*^−/−^ mice [168]. Precisely how HIF2a induces pexophagy awaits further investigation, however a favorable hypothesis is that HIF2a signaling induces the expression of an E3 ubiquitin ligase that designates peroxisomes for pexophagy.

Peroxisomes are well characterized for their roles in ROS production and scavenging; however, it also appears that an increase in cellular oxidative stress can induce pexophagy. A possible mechanism underlying oxidative stress-induced pexophagy is illustrated by work from Cheryl Walker’s group that showed peroxisomal localized mTORC1 repression that increases autophagic flux and induces pexophagy through two pathways. First, the group showed that the mTORC1 regulators, tuberous sclerosis complex 1 and 2 (TSC1 and TSC2) are localized to peroxisomes via binding to PEX19 and PEX5, respectively [170]. PEX5 itself is a ROS-sensitive protein and in the presence of ROS undergoes a conformational change that alters its TSC2-binding. With elevated ROS, PEX5-TSC2 acts as a GTPase-activating protein for Ras homologue enriched in the brain (RHEB), enabling RHEB-GTP to suppress mTORC1 and activate autophagy [170]. Second, they reported that ataxia-telangiectasia mutated (ATM) kinase is also recruited to the cytosolic face of peroxisomes in a PEX5-dependent manner in response to elevated ROS [149]. Peroxisomal ATM phosphorylates PEX5, promoting its mono-ubiquitination by the peroxisomal E3 ubiquitin ligases PEX2-PEX10-PEX12 and the induction of p62-mediated pexophagy [149]. Thus, elevated ROS both inhibits mTORC1 at the peroxisome to promote autophagic flux and specifically induces pexophagy through ubiquitin-designation. It remains unknown how RHEB specifically targets to peroxisomes, or how PEX5 retains TSC2 and ATM at the cytosolic surface instead of depositing them into the peroxisomal matrix.

A study investigating catalase provided more evidence for ROS-induced pexophagy. Lee, N. et al. found that depletion of catalase results in elevated ROS and increased pexophagy, which could be rescued with treatment of the antioxidant *N*-acetyl-*L*-cysteine [171]. Whether catalase depletion-induced pexophagy is facilitated by ATM-mediated peroxisome designation awaits further investigation. Interestingly, catalase depletion only induced pexophagy in serum-starved, but not serum-fed cells, suggesting a closer connection between starvation-induced and ROS-induced pexophagy. Further, another study reported that PEX5 depletion suppresses serum starvation-induced autophagy by downregulating TSC2 activity, resulting in active mTORC1 activity [172]. This suggests a possible role of peroxisomes as signaling nodes for autophagy via PEX5-mediated regulation of mTORC1.

Finally, a novel regulatory mechanism for non-canonical pexophagy has been demonstrated by the poly (ADP-ribose) polymerase (PARP) family members Tankyrase 1 (TNKS) and 2 (TNKS2) [173]. A proteomic analysis of TNKS and TNKS2-associated proteins found that TNKS and TNKS2 interact with PEX14 and localize to peroxisomes. Further analysis revealed that over-expression of TNKS and TNKS2 led to pexophagy induction, whilst their depletion inhibited amino-acid starvation-induced pexophagy. How TNKS and TNKS2 induce pexophagy in a ubiquitin, NBR1, and p62-independent manner is unknown, and requires further inquiry.

Thus, both yeasts and mammals have regulatory mechanisms controlling pexophagy, which connect changes in environment such as nutrient content, oxygen availability and ROS levels to peroxisome abundance. The matrix protein import cycle is also capable of regulating quality-control pexophagy in mammals to maintain a functional peroxisome population, while the existence of such mechanisms in yeasts has not been characterized.

## 7. Pexophagy in Human Health and Disease

Implications for both environment-induced and quality-control pexophagy in multiple diseases underscores the vitality of proper pexophagy for human health. The importance of quality-control pexophagy is best exemplified by a class of genetic neurodevelopmental disorders resulting from a loss of functional peroxisomes called peroxisome biogenesis disorders (PBDs). PBDs are divided into three groups based on their clinical presentations, with the most severe being Zellweger’s syndrome (ZS) that manifests in craniofacial dysmorphism, hepatomegaly and neurological abnormalities [174]. PBDs arise from mutations in 1 of 14 *PEX* genes, and it is now known that between 65% and 85% of PBDs are caused by mutations affecting pexophagy, the most common being *PEX1^G843D^* [175]. Studies of *PEX1^G843D^* mice and patient-derived fibroblasts reveal that the missense mutation results in an accumulation of ubiquitinated PEX5 in peroxisomal membranes and increased pexophagy [114]. Law, K. et al. reported that daily addition of the autophagy inhibitor, chloroquine, restored peroxisome number and functionality in *PEX1^G843D^* patient-fibroblasts [100]. Additionally, Riccio, V. et al. reported that over-expression of the pexophagy deubiquitinase, USP30, rescued peroxisome abundance in *PEX1^G843D^* patient-fibroblasts [104]. These findings point to pexophagy inhibition as a potential therapeutic strategy for PBDs. One point of confusion in PBD treatment is whether to induce or inhibit autophagy. Findings from Law, K. et al. and Riccio, V. et al. using cultured cells support that inhibition of pexophagy restores peroxisome abundance and partial function. Conversely, PEX5 mutations lead to inhibited autophagy via the TSC2-mTORC1 axis, which Eun, Y. et al. [172] suggest results in the accumulation of damaged mitochondria observed in PEX5-liver knockout mice. Thus, for PBD patients harboring PEX5 mutations, pharmacologically inducing autophagy with mTORC1 inhibitors may provide substantial benefits. As such, it is likely that the therapeutic approach for PBD patients will differ based on the afflicting mutation and tend to more personalized medicine.

A common clinical manifestation of PBD is progressive hearing loss. Intriguingly, a physiological example of ROS-induced pexophagy was reported in auditory hair cells. Overexposure to noise is known to cause oxidative stress that damages auditory hair cells and can result in hearing loss. It had been previously shown that Pejvakin, a peroxisome-associated gasdermin-family protein, protects against noise-induced oxidative stress [176], but its mechanism of action remained elusive. Defourney, J. et al. reported that sound overstimulation recruits LC3-II to peroxisomes where it binds Pejvakin via an LIR and induces pexophagy [177]. Pejvakin was found to contain two Cysteine residues, which act as ROS-sensors and are necessary for ROS-induced pexophagy in auditory hair cells. Whether Pejvakin is a bona fide pexophagy receptor or requires the aid of NBR1 and p62 to carry out pexophagy remains to be addressed. The role of Pejvakin in PBD patient-hearing loss has not been investigated.

A role for environment-induced pexophagy in severe childhood malnutrition was outlined in 2016 by Zutphen, T. et al. [151]. Severe childhood malnutrition was modeled by placing rats on a 4% low-protein diet for 4 weeks. In addition to impaired hepatic peroxisomal function, the study found that malnourished rats had diminished peroxisome abundance, concomitant with an increase in LC3-II, PEX2, NBR1 and p62 expression, suggestive of increased pexophagy. Following peroxisomal loss, mitochondrial phenotypes emerged including a loss of metabolic function resulting in decreased hepatic ATP, which could be reversed with fenofibrate supplementation that restored peroxisome abundance. These findings suggest a role for starvation-induced pexophagy in the pathogenesis of malnutrition and suggest that restoring peroxisome abundance may combat the metabolic changes seen in the liver.

Finally, a reduction in overall autophagy is associated with aging and neurodegenerative disease. Autophagy promotes longevity and increased health-span [178], and decreased autophagy is associated with the health decline observed in aging organisms [178]. It was established that peroxisome abundance increases in aging human fibroblasts and that accumulated peroxisomes displayed reduced ability to import matrix proteins including catalase [179]. This age-onset increase in peroxisome abundance and import-incompetency was mimicked in *H. polymorpha* lacking Atg1p [180], suggesting that the increase in peroxisome abundance in aged cells may be linked to decreased autophagy and pexophagy.

Notably, neurodegenerative disease is an age-onset condition correlated with declining autophagy, and a body of evidence implicates a role for peroxisomes in their pathology [181]. Brains of Alzheimer’s disease patients show altered lipid profiles including increased VLCFA in cortical regions, suggestive of defects in peroxisomal beta-oxidation [182]. Parkinson’s disease (PD) studies also show perturbed peroxisome homeostasis and depleted plasmalogens [183]. Interestingly, PBD mouse models with diminished functional peroxisomes exhibit increased alpha-synuclein oligomerization, suggesting that disrupted peroxisome homeostasis may also contribute to the characteristic alpha-synuclein inclusions associated with PD [184]. Although circumstantial, these observations implicate the age-onset accumulation of dysfunctional peroxisomes as a contributor to the pathophysiology of various neurodegenerative diseases and thus provoke the need to investigate the role of pexophagy in their pathophysiology.

## 8. Perspectives

### 8.1. Yeast vs. Mammalian Pexophagy

While pexophagy in yeasts and mammals are often viewed as analogous pathways, there are distinctive differences that call into question the conservation of the pexophagy pathway. In both yeasts and mammals, post-translational modifications are required to designate peroxisomes for pexophagy and pexophagy receptors are needed to link peroxisomes to the autophagy machinery. In mammals, pexophagy receptors NBR1 and p62 selectively target ubiquitinated peroxisomes for pexophagy, while in yeast phospho-activated pexophagy receptors Atg30p or Atg36p interact with Pex3p and Atg37p on peroxisomes to mediate pexophagy [148]. Where these two pathways may differ is in their selectivity of individual peroxisomes. The accumulation of ubiquitin on mammalian peroxisomes (Figure 3B) allows for distinction between individual peroxisomes based on their ubiquitin status. In yeasts, however, it is not known whether there are accompanying mechanism(s) that allow phosphorylated pexophagy receptors to differentiate between individual peroxisomes.

This discrepancy between yeast and mammalian pexophagy may be explained by the fact that a greater population of peroxisomes exists in mammalian cells, requiring a more stringent and selective quality-control system. Most yeasts have peroxisomes numbering under a dozen, whereas mammalian cells usually possess hundreds of peroxisomes. Therefore, while it may not be too costly for yeasts to remake most of their peroxisomes, non-selectively turning over the entire peroxisome population and reforming it is likely inefficient for mammalian cells. The selectivity of pexophagy at the individual organelle level needs to be investigated in both yeasts and mammals.

The accumulation of ubiquitinated PEX5 that is required to designate peroxisomes for pexophagy in mammals suggests a possible model for how the mammalian cell can sense a damaged peroxisome. Along with PEX5, many of the pexophagy regulators are members of the matrix protein import pathway, such as AAA-type ATPase and PEX2. Interestingly, the loss of the peroxisomal AAA-type ATPase results in the activation of pexophagy. The AAA-type ATPase is required to reset the import machinery by removing ubiquitinated PEX5 to allow continued import of matrix proteins (Figure 1). Therefore, we postulate that peroxisome matrix protein import competency is required to prevent pexophagy. In this model, peroxisomes that can no longer import matrix proteins (including all peroxisomal enzymes) are considered incompetent to carry out peroxisomal functions and are therefore degraded. 

Studies suggest that designation of yeast peroxisomes for pexophagy is ubiquitin-independent, corroborated by the fact that yeast autophagy receptors do not possess ubiquitin binding domains. Similar ubiquitin-independent pexophagy pathways involving PEX14 and PEX3 have been reported in mammalian cells. As Pex14p and Pex3p are suggested to play roles in yeast pexophagy, this raises the question of whether Pex14p/PEX14-mediated pexophagy and Pex3p/PEX3-mediated pexophagy constitute related or conserved pathways between yeast and mammalian systems. Further exploration of ubiquitin-independent pexophagy and its regulation in mammalian systems is required.

### 8.2. Peroxisomes as a Model for Selective Autophagy

In general, the selective autophagic degradation of most substrates in the mammalian cell share three common steps: (i) designation; (ii) targeting and sequestration and (iii) degradation. Where substrate degradation differs is in the specific genes that mediate and regulate these steps. One challenge in studying selective autophagy is the difficulty posed by both the biological and experimental limitations of monitoring substrate degradation. Such limitations include the activation and detection of substrate degradation, as well as the loss of cell viability upon substrate degradation. Some of these limitations are circumvented by the biological nature of peroxisomes in cultured cells, making them an excellent model for the study of selective autophagy. In cultured cells, pexophagy can be induced by either mTORC1 inhibition, or genetic manipulation of pexophagy factors. Furthermore, most cultured cells can survive in good health with depleted peroxisomes for an extended period (days), and owing to their punctate structure, peroxisomes can be readily quantified by microscopy or by biochemical techniques. Such advantages may allow researchers to address several outstanding questions in the field of selective autophagy. These include addressing the differences in the mechanisms of autophagy between organisms such as yeasts and mammals; and determining how extra- and intracellular signals may regulate selective autophagy.

Findings to-date suggest that shared pexophagy machinery is differentially regulated to coordinate either environment-induced or quality control selective autophagy. It is plausible that this holds true for other organelles such as ribosomes, lipid droplets, the ER and mitochondria. The consistency of peroxisome degradation across diverse cellular cues highlights the capacity of autophagy for selectivity and is likely mirrored in other autophagy cargoes. The exact molecular players responsible for conferring selectivity in designating, targeting and sequestering distinct autophagy cargoes for environment and quality-control induced degradation will no doubt be deciphered in the coming years.

## Figures and Tables

**Figure 1 ijms-21-00578-f001:**
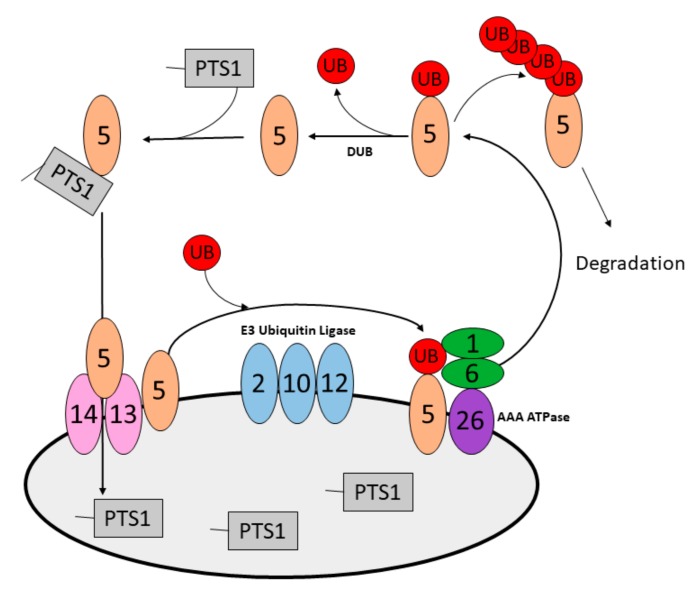
Schematic of the mammalian peroxisome matrix protein import cycle. PEX5 binds PTS1-containing proteins in the cytosol and recruits them to the peroxisome membrane through binding with PEX13-PEX14. Following the deposition of PTS1-containing proteins into the peroxisome lumen, PEX5 is ubiquitinated by the RING-like E3 ubiquitin ligases PEX2, PEX10, PEX12. Ubiquitinated PEX5 is removed from the membrane by the AAA-type ATPase complex, PEX1-PEX6-PEX26. In the cytosol, PEX5 is either deubiquitinated to enable another round of import or polyubiquitinated and proteasomally degraded.

**Figure 2 ijms-21-00578-f002:**
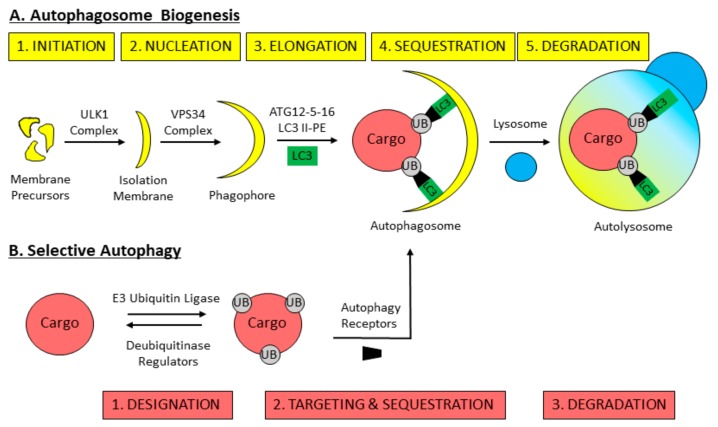
Schematic of mammalian selective autophagy. (**A**) Stages of autophagosome biogenesis and degradation. The ULK1 complex drives autophagy initiation to form the isolation membrane from membrane precursors. Nucleation of the isolation membrane by the VPS34 complex forms a cup-shaped membrane termed the phagophore. The phagophore elongates by the ubiquitin-like conjugation systems, ATG12-5-16 and LC3-II-phosphatidylethanolamine (PE), to form the autophagosome. The autophagosome sequesters cargoes within and undergoes fusion with a lysosome to form an autolysosome, where degradation occurs. (**B**) Selective autophagy cargoes are often marked for degradation by the addition of ubiquitin on their outer surface by E3 ubiquitin ligases. Ubiquitin designation signals can be dissipated from cargoes by deubiquitinases and regulators. Designated cargoes are targeted to the autophagosome through binding with autophagy receptors that interact with LC3-II on the autophagosome. Cargoes sequestered inside autophagosomes through binding with autophagy receptors are degraded in an autolysosome.

**Figure 3 ijms-21-00578-f003:**
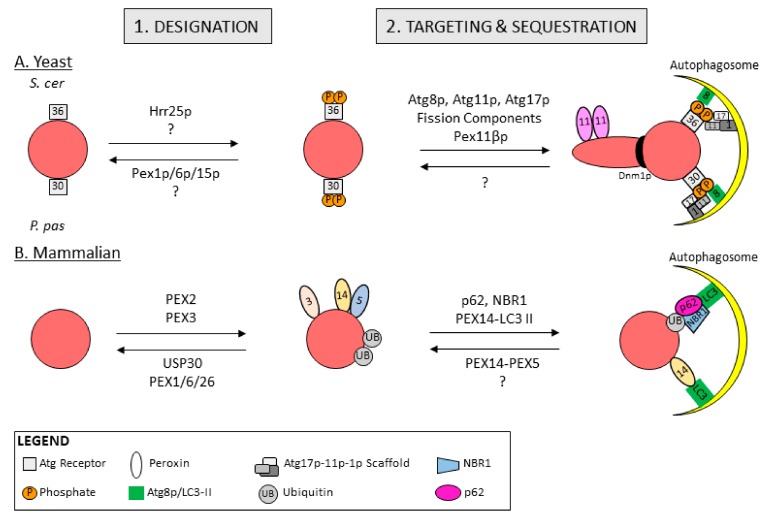
Schematic of pexophagy mechanisms of designation, targeting and sequestration in yeast and mammalian systems. (**A**) *S. cerevisiae* autophagy receptor Atg36p and *P. pastoris* autophagy receptor Atg30p are phosphorylated by Hrr25p and an unknown kinase to designate peroxisomes for autophagy. Phosphorylated Atg36p and Atg30p interact with scaffolding proteins Atg11p, Atg17p and Atg1p that target them to autophagosomes. Interactions between phosphorylated Atg36p, Atg30p and Atg8p further sequester designated peroxisomes within autophagosomes. Pex11βp-mediated fission further aids peroxisome sequestration within the autophagosome. (**B**) Peroxisome membrane proteins are ubiquitinated by the E3 ubiquitin ligase, PEX2, to designate peroxisomes for pexophagy. Ubiquitinated peroxisome membrane proteins are removed from peroxisomes by the AAA-type ATPase PEX1-PEX6-PEX26 and the deubiquitinase USP30 to prevent pexophagy. Increasing expression of PEX3 on peroxisome membranes may also designate them for pexophagy. Ubiquitinated peroxisomes are targeted to autophagosomes through interactions with the autophagy receptors NBR1 and p62, which facilitate sequestration within autophagosomes through binding with LC3-II. Peroxisomes are also targeted and sequestered within autophagosomes when LC3-II out-competes PEX5 for binding to PEX14.

**Figure 4 ijms-21-00578-f004:**
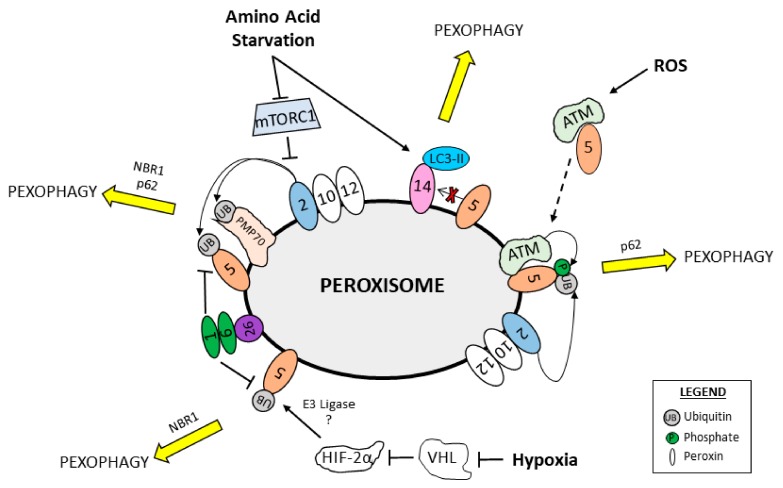
Regulation of mammalian pexophagy. Schematic of ubiquitin-independent and ubiquitin-dependent pexophagy resulting from intracellular and extracellular cues. Amino acid starvation relieves mTORC1 inhibition of the peroxisomal E3 ubiquitin ligase PEX2, allowing PEX2 to ubiquitinate peroxisomal proteins PMP70 and PEX5 and facilitating NBR1 and p62-mediated pexophagy. LC3-II-PEX14 interactions promote pexophagy in an amino acid starvation-dependent mechanism. Increased reactive oxygen species (ROS) levels result in ATM–PEX5 interactions at the peroxisome that promote PEX5 phosphorylation by ATM and ubiquitination by PEX2, to facilitate p62-mediated pexophagy. Hypoxia relieves VHL-mediated inhibition of HIF-2a, resulting in PEX5 ubiquitination by an unknown E3 ubiquitin ligase and NBR1-mediated pexophagy. The peroxisomal AAA-type ATPase complex PEX1-PEX6-PEX26 opposes PEX5-ubiquitination and pexophagy.

**Table 1 ijms-21-00578-t001:** Proteins involved in pexophagy. Table 1 was prepared to reflect the steps of pexophagy: autophagosome biogenesis, peroxisome designation, targeting and sequestration and degradation (as illustrated in Figure 2). The autophagosome biogenesis genes were adapted from Reference [64]. The pexophagy factors were manually curated from literature. Here, slash nomenclature denotes isoforms while commas denote subfamilies. Yeast proteins are named followed by the mammalian protein homologue in brackets in the ‘Function/Role’ column, when applicable.

Proteins Involved in Pexophagy				
	Yeast	Mammalian	Function/Role	Reference
Autophagosome Biogenesis				
**Atg1p/ULK Complex**	Atg1p	ULK1/2	Ser/Thr kinase	[61]
	Atg13p	ATG13	Regulatory subunit of Atg1 (ULK1/2) complex	[63,65]
	Atg11p		Scaffold protein in pexophagic PAS	[68]
	Atg17p	FIP200	In complex with Atg29p-Atg31p (ATG13-ATG101)	[64,65,66]
	Atg29p		In complex with Atg17p and Atg31p	[62]
	Atg31p		In complex with Atg17p and Atg29p	[68]
		ATG101	In complex with ATG13-FIP200	[67]
**Atg9p/ATG9 membrane cycling**	Atg2p	ATG2	Interacts with Atg18p (WIPI1/2)	[76]
	Atg9p	ATG9A/B	Transmembrane protein; supplies membrane for autophagosome in vesicles	[87]
	Atg18p	WIPI1/2	PtdIns3P-binding protein at autophagosome-ER contact sites	[76]
**PtdIns3K complex**	Vps34p	VPS34	PtdIns 3-kinase	[74,75]
	Vps15p	VPS15	Ser-Thr kinase	[74,75]
	Atg6p	BECN1	Component of PtdIns3K complex I	[74,75]
	Atg14p	ATG14	Component of PtdIns3K complex II	[74,75]
**Atg8p/LC3 Ubiquitin-like conjugation system**	Atg8p	LC3A/B/C, GABARAP, GABARAPL1/2	Ubiquitin-like protein conjugated to PE	[80,82,83,84,85]
	Atg7p	ATG7	E1-like enzyme	[80]
	Atg3p	ATG3	E2-like enzyme	[80]
	Atg4p	ATG4A/B/C/D	Cysteine protease that cleaves Atg8 (LC3)	[81]
**Atg12p/ATG12 Ubiquitin-like conjugation system**	Atg12p	ATG12	Ubiquitin-like protein	[77]
	Atg7p	ATG7	E1-like enzyme	[77]
	Atg10p	ATG10	E2-like enzyme	[78]
	Atg16p	ATG16L1	Interacts with Atg5 and Atg12 (ATG5, ATG12) to aid ubiquitin-like conjugation	[79]
	Atg5p	ATG5	Substrate of Atg12 (ATG12)-conjugation	[77]
**Peroxisome Designation**				
**AAA-type ATPase Complex**	Pex1p	PEX1	In complex with Pex6-Pex15 (PEX6-PEX26); Pex5 (PEX5) receptor recycling; defects signal pexophagy	[99,100]
	Pex6p	PEX6	In complex with Pex1-Pex15 (PEX1-PEX26)	[99,100]
	Pex15p	PEX26	In complex with Pex1-Pex6 (PEX1-PEX6)	[99,100]
**E3 Ubiquitin Ligase**		PEX2	In complex with PEX10-PEX12; ubiquitinates PEX5 and PMP70 to signal mammalian pexophagy	[101]
**Ubiquitin Targets**		PEX5	Matrix protein import receptor; accumulated PEX5-UBB signals pexophagy	[99,100,101,102]
		PMP70	PMP; accumulated PMP70-UBB signals pexophagy	[101]
**Deubiquitinase**		USP30	DUB; removes ubiquitin from PEX5 and PMP70 to oppose pexophagy	[103,104]
**Unknown**	Pex3p	PEX3	Biogenesis factor; loss signals yeast pexophagy, over-expression signals mammalian pexophagy	[105,106]
	Pex14p	PEX14	Defects signal yeast pexophagy; potential signal for mammalian pexophagy	[99,107]
**Peroxisome Targeting and Sequestration**				
**Pexophagy Receptors**	Atg30p		Links peroxisomes to autophagy machinery; *P. pastoris*	[108]
	Atg36p		Links peroxisomes to autophagy machinery; *S. cerevisiae*	[109,110]
		NBR1	Links peroxisomes to autophagy machinery; primary receptor	[111,112]
		p62	Links peroxisomes to autophagy machinery; enhances NBR1-mediated pexophagy	[111,112]
**Receptor Ligands**	Atg37p	ACBD5	Tethers Atg30p to peroxisomes in *P. pastoris;* regulates formation of receptor protein complex	[113]
	Pex3p		Binds and tethers Atg30p to peroxisomes in *P. pastoris;* and Atg36p to peroxisomes in *S. cerevisiae*	[114]
		UBB	NBR1 and p62 bind ubiquitin via UBA	[111]
**Kinases**	Hrr25p		Phosphorylates Atg30p in *P. pastoris* and Atg36p in *S. cerevisiae*; promotes pexophagy	[113,114]
**Fission Machinery**	Dnm1p		Fission machinery; pinches off peroxisomes for yeast pexophagy	[115,116]
	Vps1p			
	Fis1p			
	Mffp			
	Pex11p			
**Peroxisome Degradation**				
**Fusion Machinery**	Atg24p		PtdIns3P-binding protein; required for vacuolar fusion in pexophagy	[117]
**Cytosolic Peroxin Degradation**	Atg11p		Facilitates degradation of Pex5p and Pex7p during pexophagy	[118]

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
