# Peer review of "Pexophagy: A Model for Selective Autophagy"

_ijms, 2020, doi:10.3390/ijms21020578_

Round 1

Reviewer 1 Report

This is a very balanced, complete and authoritative review. The authors discuss pexophagy in an inspiring way, they draw the big picture and are not shy of going into the details of mechanism.

If there was one major point, I as a reader would expect a little more guidance as to the reliability of the information given in the review. All results are presented with the same intensity. In how far are the conclusions that can be drawn from mammalian models complementary to yeasts? Which data are reliable in the eyes of the authors, which less so?

All my other comments are very minor points that may help to further improve an already great review:

Line 2 (title) and abstract: The ”model”-aspect of the review does not become clear. Only in line 246, this question it taken up. I understand that pexophagy is a “case” or an “example” of environment-induced and of quality-control autophagy, but why and how it can be used to model eg. mitophagy using pexophagy is not clear to me after reading the full review. I suggest to change the title or to explain the model-function of pexophagy in the text.

Line 27: consider deleting “colleague,”

Line 31: “peroxisomes … express” – given that peroxisomes cannot really express gene, this may be misleading

Line 37: sterol precursor synthesis in peroxisomes is highly contentious, should be mentioned

Line 61-70 (paragraph): Mammalian peroxisomes is discussed here, authors should delete or clearly mark references to yeast

Line 91 and many other occurrences: “AAA ATPase” as the first A in AAA refers to ATPase, I suggest to change to AAA, ATPase complex, or AAA-type ATPase

Line 138, Table 1, and many occurrences in the text: Yeast protein nomenclature is not used consistently throughout the text (with/without “p”)

Table 1 and many occurrences in the text: The “slash-nomenclature” (eg. RB1CC1/FIP20) is not used consistently. It would be good to define (generally or in each case), if such an expression refers to homologs, to alternative names for the same, or to (non-homologous) components of a protein complex

Line 365 and 373: What does “gross” (ubiquitination) refer to?

Line 367 Explain how abundance of PMP70 and the hypothesis of several PEX2 membrane protein targets are related.

Line 532 “non-carbon fermentable” - please check! “non-fermentable carbon” may be meant here

Line 536 and 536: “… are reported to play … “ – a reference is required here.

Line 536 and 537: are RGT2 and SNF3 themselves related? What is the link between the four proteins mentioned?

Line 550: The argument is not clear here: A potential regulator (palmitoyl-CoA) does not have to be a substrate, or?

Line 552: “stimulus”, consider changing to “stimuli”

Line 555: “low protein levels” – of what?

Line 556: how can there be “ex vivo” starvation?

Line 558 and 559: “restoring” – consider: “returning to” - not clear which basal expression levels are restored by which factor.

Line 562: sentence not clear

Line 564: “For peroxisomes role in oxidative metabolism” – expression not clear

Line 564 and other occurrences: reference to author names in the text is not consistent: first names should be spelled out, abbreviated, or left out (compare line 27 or line 577)

Line 566 delete “-“

Line 569: “Hipel”, change to “Hippel”

Line 571: introduce “EPAS”

Line 584: “suppress” (spelling)

Line 604 to 608: Are TNSK2 and TNK2 the same?

Line 631: “PBD’s” change to “PBDs”

Line 634-636: It somewhat implied that PEX1 G843D is the most common mutation affecting pexophagy – Is that so?

Line 720: explain “UBKo”

Line 738 (abbreviations): “associating”, consider “associated”; delete “-“ in “PI-phosphate”

Line 749: Journal abbreviation is not used consistently in reference list (compare line 830 and line 846)

Line 762: Check Journal abbreviation “Curr. Biol. CB”

Line 775: Check Journal abbreviation

Line 787: Reference incomplete

Line 901: Check page range

Line 963: Check Journal abbreviation “ … Devoted …”

Line 967 and 972 and 1029: Check Journal abbreviation “ … Cph. Den.”

Line 1006: Reference incomplete

Line 1090: Delete “( Pt 10)”

Line 1172: Check Journal abbreviation “ … TEM”

Line 1175: Check Journal abbreviation “ … (Berl.)”

Line 1179: Check Journal abbreviation “ … Camb. Mass.”

Author Response

This is a very balanced, complete and authoritative review. The authors discuss pexophagy in an inspiring way, they draw the big picture and are not shy of going into the details of mechanism. If there was one major point, I as a reader would expect a little more guidance as to the reliability of the information given in the review. All results are presented with the same intensity. In how far are the conclusions that can be drawn from mammalian models complementary to yeasts? Which data are reliable in the eyes of the authors, which less so?

First, we like to thank the reviewer for taking the time to give us feedback on our review manuscript. We feel that the changes recommended by both reviewer strengthen our manuscript. Please find below summary of our changes and comments each of the reviewer's concerns.

In the question of giving guidance, we struggled with this because we feel that the current state of knowledge of pexophagy is such that there is very little contrasting data. This made it difficult to give an unbiased critique of the various results. To partly alleviate the reviewers concern we expanded our conclusion and perspectives section to include a paragraph on how and why yeast vs. mammalian systems differ in their pexophagy pathway.

All my other comments are very minor points that may help to further improve an already great review: Line 2 (title) and abstract: The “model”-aspect of the review does not become clear. Only in line 246, this question it taken up. I understand that pexophagy is a “case” or an “example” of environment-induced and of quality-control autophagy, but why and how it can be used to model eg. Mitophagy using pexophagy is not clear to me after reading the full review. I suggest changing the title or to explain the model-function of pexophagy in the text.

In response to this comment, we have expanded our conclusion and perspectives section to include a paragraph detailing why pexophagy is a good model of selective autophagy.

Line 27: consider deleting “colleague,”

DONE

Line 31: “peroxisomes … express” – given that peroxisomes cannot really express gene, this may be misleading  

We have changed the sentence to reflect that the peroxisomes ubiquitously harbour a set of peroxin proteins, not genes.

Line 37: sterol precursor synthesis in peroxisomes is highly contentious, should be mentioned

In response to this comment we have changed the sentence to soften the claim.

Line 61-70 (paragraph): Mammalian peroxisomes is discussed here, authors should delete or clearly mark references to yeast

            We have deleted the yeast references.

Line 91 and many other occurrences: “AAA ATPase” as the first A in AAA refers to ATPase, I suggest changing to AAA, ATPase complex, or AAA-type ATPase

We have changed these occurrences to AAA-type ATPase

Line 138, Table 1, and many occurrences in the text: Yeast protein nomenclature is not used consistently throughout the text (with/without “p”)

            We have changed these occurrences to “p” for consistent nomenclature.

Table 1 and many occurrences in the text: The “slash-nomenclature” (eg. RB1CC1/FIP20) is not used consistently. It would be good to define (generally or in each case), if such an expression refers to homologs, to alternative names for the same, or to (non-homologous) components of a protein complex

We have changed the slash-nomenclature to reflect homologs and defined it as such in the text and Table I.

Line 365 and 373: What does “gross” (ubiquitination) refer to?

We have changed these instances to mass ubiquitination which is referring to large-scale ubiquitination of peroxisomes.

Line 367 Explain how abundance of PMP70 and the hypothesis of several PEX2 membrane protein targets are related.

We have simplified this concept by inserting the sentence: Whether PEX5 and PMP70 are the only proteins selectively ubiquitinated is not known, however, given the non-selectivity of other autophagy E3 ubiquitin ligases, PEX2 likely ubiquitinates additional peroxisomal membrane proteins during amino acid starvation

Line 532 “non-carbon fermentable” - please check! “non-fermentable carbon” may be meant here

            We have changed to non-fermentable carbon.

Line 536 and 536: “… are reported to play … “ – a reference is required here.

            Reference has been included.

Line 536 and 537: are RGT2 and SNF3 themselves related? What is the link between the four proteins mentioned?

We have changed this paragraph to better explain the proteins. The link between the four proteins is that they are all involved in glucose sensing. “In S. cerevisiae, the cell-surface glucose sensor composed of the G-protein-coupled receptor Gpr1p and G-protein Gpa2p, which are involved in cAMP-signalling, were shown to regulate glucose-induced pexophagy [163]. Additionally, the high and low affinity glucose sensors Snf3p and Rgt2p, respectively, are reported to play a role in glucose-sensing to initiate pexophagy [168]. Homologs of Rgt2p and Snf3p function similarly in P. pastoris (Gss1p) [163] and H. polymorpha (Gcr1p) [165].”           

Line 550: The argument is not clear here: A potential regulator (palmitoyl-CoA) does not have to be a substrate, or?

We have clarified the argument by adjusting the paragraph to: In vitro work suggests that palmitoyl-CoA and Atg30p compete for the same binding site on the N-terminal cytosolic domain of Atg37p, although it does not affect Atg37p-Pex3p binding [132]. These findings prompt speculation as to whether increasing concentrations of palmitoyl-CoA may inhibit pexophagy in certain conditions.

Line 552: “stimulus”, consider changing to “stimuli”     

            Done.

Line 555: “low protein levels” – of what?

Changed sentence to clarify: In basal conditions, PEX2 is rapidly turned over resulting in overall low expression of PEX2.

Line 556: how can there be “ex vivo” starvation?

            Changed to in vitro.

Line 558 and 559: “restoring” – consider: “returning to” - not clear which basal expression levels are restored by which factor.

            Done.

Line 562: sentence not clear

Changed to: As such, it is tempting to speculate that during basal conditions mTORC1 promotes the proteasomal degradation of PEX2. Whereas, upon starvation or other conditions that inhibit mTORC1, PEX2 is stabilized and can promote the ubiquitination of peroxisomal proteins to designate peroxisomes for pexophagy.

Line 564: “For peroxisomes role in oxidative metabolism” – expression not clear

            Done.

Line 564 and other occurrences: reference to author names in the text is not consistent: first names should be spelled out, abbreviated, or left out (compare line 27 or line 577)

Done.

Line 566 delete “-“

            Done.

Line 569: “Hipel”, change to “Hippel”

            Done.

Line 571: introduce “EPAS”

            Changed to VHL-/- for simplicity.

Line 584: “suppress” (spelling)

            Done.

Line 604 to 608: Are TNSK2 and TNK2 the same?

This was a typo…they are 2 different proteins. The paragraph has been adjusted to: family members tankyrase 1 (TNKS) and 2 (TNKS2)

Line 631: “PBD’s” change to “PBDs”

            Done.

Line 634-636: It somewhat implied that PEX1 G843D is the most common mutation affecting pexophagy – Is that so?

            Yes, see lines 636-7.

Line 720: explain “UBKo”

            Changed to “a ubiquitinated PMP34 construct”

Line 738 (abbreviations): “associating”, consider “associated”; delete “-“ in “PI-phosphate”

            Done.

Line 749: Journal abbreviation is not used consistently in reference list (compare line 830 and line 846)

            All changed to the same abbreviation.

Line 762: Check Journal abbreviation “Curr. Biol. CB”

            Updated to correct abbreviation.

Line 775: Check Journal abbreviation

            Done.

Line 787: Reference incomplete

Updated to complete reference.

Line 901: Check page range

            Updated to correct page range.

Line 963: Check Journal abbreviation “ … Devoted …”

Updated to correct abbreviation.

Line 967 and 972 and 1029: Check Journal abbreviation “ … Cph. Den.”

            Updated to correct abbreviation.

Line 1006: Reference incomplete

Updated to complete reference.

Line 1090: Delete “( Pt 10)”

Done.

Line 1172: Check Journal abbreviation “ … TEM”

            Updated to correct abbreviation.

Line 1175: Check Journal abbreviation “ … (Berl.)”

            Updated to correct abbreviation.

Line 1179: Check Journal abbreviation “ … Camb. Mass.”

            Updated to correct abbreviation.

Reviewer 2 Report

Dear Editor,

I read the review paper by Germain and Kim, which is interesting, well-written and balanced in its parts. My only concerns regards the choice of the references, which are several and dated. Secondly, a method section describing in detail how the data searching process was performed is needed.  I would like the Authors revise their manuscript accordingly with this suggestion.

Author Response

I read the review paper by Germain and Kim, which is interesting, well-written and balanced in its parts. My only concern regards the choice of the references, which are several and dated. Secondly, a method section describing in detail how the data searching process was performed is needed. I would like the Authors to revise the manuscript accordingly with this suggestion.

We thank the reviewers for the helpful comments. Please find below a summary of our changes and comments to the reviewers concerns.

Regarding the references, we have revised a few but feel that those included are necessary. Some of the reference may appear dated as we have referenced primary sources.  

We have updated our Table I caption to explain its method of creation:

“Table 1 was prepared to reflect the steps of pexophagy: autophagosome biogenesis, peroxisome designation, targeting and sequestration, and degradation (as illustrated in Fig. 2). The autophagosome biogenesis genes were adapted from Ref. 64. The pexophagy factors were manually curated from literature. Here, slash nomenclature denotes isoforms while commas denote subfamilies. Yeast proteins are named followed by the mammalian protein homologue in brackets in the ‘Function/Role’ column, when applicable.”